# Investigation of Forces and Moments during Orthodontic Tooth Intrusion Using Robot Orthodontic Measurement and Simulation System (ROSS)

**DOI:** 10.3390/bioengineering10121356

**Published:** 2023-11-25

**Authors:** Corinna L. Seidel, Julian Lipp, Benedikt Dotzer, Mila Janjic Rankovic, Matthias Mertmann, Andrea Wichelhaus, Hisham Sabbagh

**Affiliations:** Department of Orthodontics and Dentofacial Orthopedics, University Hospital, LMU Munich, Goethestrasse 70, 80336 Munich, Germany; julian.lipp@med.uni-muenchen.de (J.L.); mila.janjic@med.uni-muenchen.de (M.J.R.); matthias.mertmann.extern@med.uni-muenchen.de (M.M.); kfo.sekretariat@med.uni-muenchen.de (A.W.); hisham.sabbagh@med.uni-muenchen.de (H.S.)

**Keywords:** nickel-titanium, NiTi archwires, initial levelling and aligning, intrusion, intrusion steps, orthodontic materials, biomechanical orthodontic simulation, orthodontic tooth movement, force control

## Abstract

The Robot Orthodontic Measurement and Simulation System (ROSS) is a novel biomechanical, dynamic, self-regulating setup for the simulation of tooth movement. The intrusion of the front teeth with forces greater than 0.5 N poses a risk for orthodontic-induced inflammatory root resorption (OIIRR). The aim was to investigate forces and moments during simulated tooth intrusion using ROSS. Five specimens of sixteen unmodified NiTi archwires and seven NiTi archwires with intrusion steps from different manufacturers (Forestadent, Ormco, Dentsply Sirona) with a 0.012″/0.014″/0.016″ wire dimension were tested. Overall, a higher wire dimension correlated with greater intrusive forces F_z_ (0.012″: 0.561–0.690 N; 0.014″: 0.996–1.321 N; 0.016″: 1.44–2.254 N) and protruding moments M_x_ (0.012″: −2.65 to −3.922 Nmm; 0.014″: −4.753 to −7.384 Nmm; 0.016″: −5.556 to −11.466 Nmm) during the simulated intrusion of a 1.6 mm-extruded upper incisor. However, the ‘intrusion efficiency’ parameter was greater for smaller wire dimensions. Modification with intrusion steps led to an overcompensation of the intrusion distance; however, it led to a severe increase in F_z_ and M_x_, e.g., the Sentalloy 0.016″ medium (Dentsply Sirona) exerted 2.891 N and −19.437 Nmm. To reduce the risk for OIIRR, 0.014″ NiTi archwires can be applied for initial aligning (without vertical challenges), and intrusion steps for the vertical levelling of extruded teeth should be bent in the initial archwire, i.e., 0.012″ NiTi.

## 1. Introduction

The orthodontic intrusion of the front teeth with fixed appliances can be necessary to correct deep bites [1]; yet, orthodontically induced inflammatory root resorptions (OIIRRs) are the most undesired side effect [2] with an incidence up to 90% during treatment with fixed appliances [3,4,5,6,7]. A recent systematic review summarized how the incidence and severity of OIIRR correlate with high forces, tooth intrusion, the application of torque, the retraction of the front teeth as well as orthodontic treatment with enhanced duration, premolar extractions, or long distances of tooth movement [2]. Furthermore, OIIRRs were shown to correlate with genetic predisposition [8] and the adaptability of tissue [9,10,11,12]. Moreover, the maxillary front teeth are susceptible to OIIRRs [13,14,15,16]. Adequate forces and moments during orthodontic tooth movement (OTM) have to be differentiated clearly and are dependent on several factors like (I) tooth type and the size of the root surface (incisors, canines, premolars, molars), (II) the type of tooth movement (tipping, translational, torque), (III) the duration of force application (continuous, intermittent as well as IV) and age and health condition [1]. While axial and short-term forces of up to 300 Newtons (N) can be compensated, forces of solely 1 N can lead to OIIRR if applied non-axially and for a longer duration [17]. Reitan et al. defined forces for each tooth type depending on the type of tooth movement varying between 20 centi Newtons (cN) for the extrusion of the front teeth and 245 cN for the translational movement of canines toward the end of treatment [18]. Further, a moderate force application of 10–50 cN was shown to be more effective, while leading to fewer side effects [19,20,21]. Intrusive forces induce the apical compression of the periodontal ligament, which is associated with enhanced risk for OIIRR. Faltin et al. showed that continuous forces of 50 cN applied during the intrusion of premolars can lead to OIIRR [22]. In accordance with histological studies [18,19,20,21,22,23], orthodontic forces used for tooth intrusion should range between 0.1 and 0.3 N [1]. Orthodontic tooth intrusion can be performed during levelling and aligning with standard nickel-titanium (NiTi) archwires [24] or with modified NiTi archwires and intrusion steps [25]. In daily practice, orthodontists can modify wires using cold forming with pliers or heat treatment using a direct electric resistance heater, such as the Memory-Maker^TM^ (Forestadent, Pforzheim, Germany) [26]; however, the quality of the steps depends on the individual manual and technical accuracy. Heat treatment with a furnace using defined annealing temperatures and annealing durations for the thermal programming of NiTi archwires is preferred to ensure comparable experimental conditions [27].

Forces and moments during OTM have been intensively studied, e.g., using the three-point bending test [27,28,29,30]. However, due to its limited transferability to clinical in vivo situations, the three-point bending test is mainly used for orthodontic material science [31]. Hence, several new biomechanical simulation setups have been developed in recent decades. Bourauel et al. presented the Orthodontic Measurement and Simulation System (OMSS), a software-based setup with two force–moment-sensors and the ability to investigate force-controlled orthodontic tooth movements three-dimensionally [32], which has been used to address several research questions [33,34]. Friedrich et al. first described an in vivo setup to investigate forces exerted by orthodontic appliances using a force–moment-sensor adapted to a separable bracket, though without reproducible results and limited dimensionality [35]. Wichelhaus and Sander et al. developed a biomechanical three-dimensional test setup with two force–moment sensors and a thermal chamber, that was able to analyse forces and moments considering four degrees of freedom [36]. Fuck et al. designed the Robotic Measurement System (RMS), which was able to analyse forces and moments during initial orthodontic tooth movement regarding six degrees of freedom [37]. The Orthodontic Simulator (OSIM) was invented by Badawi et al. with 14 force–moment sensors for each tooth; yet, the use of several sensors led to feedback effects, and solely one-dimensional deviations could be analysed [38]. The Orthodontic Force Tester (OFT) was implemented by Chen et al. using two sensors to measure the forces and moments affecting adjacent teeth during space closure; however, this setup also faced problems with side effects, e.g., activation in the distal direction led to unwanted tooth deflections and, therefore, potentially distorted measurement forces [39,40]. Perrey et al. used OMSS to analyse the impact of different bracket types and archwire qualities on forces and moments during OTM [41,42]. Further, Finite Element Methods (FEM) were established using the digital simulation of OTM without mechanical test setups [43,44]. The conventional mechanical test setups were compared to FEM by Hayashi et al., presenting comparable results for both techniques [45]. Lately, the Robot Orthodontic Measurement and Simulation System (ROSS), consisting of a precision robot, a force–torque sensor, a thermal chamber, and a test tooth, was introduced to investigate the forces and moments exerted during intrusive, rotational, and angular tooth movement [24]. 

The aim of this study was to investigate forces and moments during the simulated intrusion of extruded front teeth using ROSS exerted by sixteen commercially available NiTi archwires with different wire dimensions and the following two therapeutic approaches: vertical levelling using (I) standard NiTi archwires and (II) modified NiTi archwires with intrusion steps. To ensure comparability, we developed an intrusion step shape setting tool using Computer-Aided Design (CAD). The overarching goal was to experimentally identify wires exerting the most adequate forces and moments for the orthodontic intrusion of the front teeth to reduce the risk of OIIRR.

## 2. Materials and Methods

### 2.1. Robot Orthodontic Measurement and Simulation System 

The Robot Orthodontic Measurement and Simulation System (ROSS) was developed in the Biomechanics Laboratory of the Department of Orthodontics and Dentofacial Orthopedics of the LMU University Hospital [24]. It consists of a precision robot KUKA KR 5-sixx R650 (KUKA Roboter GmbH, Gersthofen, Germany), an FTS Nano 17 SI-12-0.12 force–torque sensor (ATI Industrial Automation, Apex, NC, USA), a test tooth on a high-strength titanium alloy Kavo Typodont model adapted to SAM^®^ Axiosplit^®^ mounting plates (SAM Präzisionstechnik GmbH, Gauting, Germany) (Figure 1a) and a thermal chamber with a temperature controller REX-C100 PID (RKC Instrument Inc., Japan, Ibaraki), ensuring test conditions of 37.0 ± 0.5 °C [24]. An upper central incisor (11) was modified with bonded active self-ligating, and a straight wire bracket with an MBT prescription and 0.022″ slot (Bioquick, Forestadent, Germany, Pforzheim) applying passive pre-bent 0.021″ × 0.025″ steel wire for the positioning of the bracket. The spatial coordinate system was used for the test sensor; hence, the x-, y-, and z-axis presented the mesiodistal, orovestibular, and vertical axis of the bracket slot. Since the intrusion of 11 was investigated, mainly F_z_ (positive values = intrusive forces; negative values = extrusive forces) and M_x_ (positive values = retruding moments; negative values = protruding moments) were of interest, while data in all translational and rotational directions were collected.

### 2.2. Experimental Setup and Measurement Simulation 

The test setup comprised the following two stages: Position the test tooth adapted to the force–torque sensor in the ideal position according to the KAVO typodont model using a silicone impression. The deflection of the test tooth in the extruded, pathologic starting position is obtained (extrusion distance: 1.6 mm), and the test archwire is inserted (Figure 1a).The automatic and gradual transfer of the test tooth using force control (pre-defined in LabView) toward its final position is achieved, which is defined as forces and moments approaching zero (Figure 1b). As the measured intrusion distance did not reach 1.6 mm for each investigated wire, a reference distance was defined as 0.8 mm for statistical evaluation to ensure the data points for each wire and the comparability of data. The LabView software (LabView 2012 Version 12.03f3) by the manufacturer National Instruments (NI, Austin, TX, USA) was used for closed-loop force control using virtual instruments (VI) and processes based on mathematical algorithms. The force–torque sensor measured forces and moments, continuously adding up to thousands of data points for each sample. Afterward, the test wire was removed, and the sensor was reset for the next test series.

Five wire samples from each of the following NiTi archwires were tested in independent test cycles: 0.012″/0.014″/0.016″ Biostarter (Forestadent, Germany, Pforzheim), 0.012″/0.014″/0.016″ Align XF (Ormco, Brea, CA, USA), 0.012″/0.014″/0.016″ Copperloy (Dentsply Sirona, York, PA, USA), 0.014″/0.016″ Sentalloy Light (Dentsply Sirona, York, PA, USA); 0.012″/0.014″/0.016″ Sentalloy Medium (Dentsply Sirona, York, PA, USA), 0.014″/0.016″ Sentalloy Heavy (Dentsply Sirona, York, PA, USA). Moreover, five additional wire samples of all Sentalloy wires were modified with intrusion steps and tested. To ensure the reproducibility of the intrusion step, an intrusion step template was designed using CAD and manufactured using stainless steel (Figure 1c,d). The intrusion step was 1.2 mm high and 9 mm wide in accordance with the width of the test tooth. The heat treatment of the NiTi wire was performed using the Heraeus K 750/1 furnace (Heraeus GmbH, Germany, Hanau) with a chamber temperature of 550 °C and an annealing duration of 16 min. To ensure the exact annealing temperature inside the tool, a Ni-CrNi thermocouple was applied to the tool in very close proximity to the wire. 

### 2.3. Statistics

Descriptive statistics were generated using SPSS Statistics 26 (IBM, USA, NY, Armonk) and Microsoft Excel 2016 (Microsoft Corporation, USA, Washington, Redmond). Graphical presentation was performed using OriginProm Version 2020 (Origin-Lab Corporation, Northampton, MA, USA). A power analysis was performed to verify an adequate, quantitative sample size. Variance analyses were performed using the Kruskal–Wallis test. Differences were considered significant with *p*-values ≤ 0.05.

## 3. Results

### 3.1. Comparison of Different Wire Dimensions of NiTi Archwires by the Same Manufacturer without Intrusion Steps

As for Forestadent wires, the initial intrusive forces (F_z_) increased with larger wire dimensions with a peak of 1.578 N for Biostarter 0.016″ archwires (Table 1). Accordingly, higher wire dimensions presented greater initial protruding moments (M_x_) with a maximum of −7.261 Nmm for Biostarter 0.016″ archwires (Table 1). Similar intrusion distances were seen for Biostarter 0.012″ and 0.014″ archwires, while 0.016″ archwires reached higher distances of 1.281 mm. 

With regard to Ormco wires, increasing intrusive forces, protruding moments, and intrusion distances were found for AlignXF archwires with an increasing wire diameter (Table 1). AlignXF 0.016″ demonstrated the second-highest intrusive forces (F_z_ = 2.180 N) and the highest protruding moments (M_x_ = −11.466 Nmm) in comparison to other manufacturers. The differences between the smallest and greatest wire dimensions were 1.490 N for intrusive forces, 7.544 Nmm for protruding moments and 0.156 mm for intrusion distances, which is in accordance with other manufacturers.

Regarding Dentsply Sirona wires, initial intrusive forces, protruding moments, and intrusion distances were greater with higher wire dimensions (Table 1). Copperloy 0.016″ archwires presented higher intrusive forces (F_z_ = 1.853 N), protruding moments (M_x_ = −5.556 Nmm), and intrusion distances (z = 1.031 mm) compared to Copperloy 0.014″. Sentalloy 0.012″ medium archwires reached mean initial forces F_z_ of 0.621 N, while Sentalloy 0.016″ heavy showed the highest initial intrusive forces of 2.254 N, which is three times higher compared to the Sentalloy 0.012″ medium. Similarly, the protruding moments of Sentalloy 0.016″ heavy archwires reached almost three times higher values (M_x_ = −10,047 Nmm) compared to the Sentalloy 0.012″ medium. The intrusion distances were enhanced with a higher wire dimension except for Sentalloy 0.012″ medium with greater distances than Sentalloy 0.014″ light. The maximum difference between the intrusion distances was 0.240 mm, which is similar to other manufacturers.

Taken together, Biostarter 0.016″ (Forestadent) archwires reached the highest intrusion distances, while the initial intrusive forces were second lowest compared to 0.016″ archwires by other manufacturers (Figure 2). Biostarter 0.012″ archwires presented low mean initial forces (F_z_ = 0.561 N) and moments (M_x_ = −2.649 Nmm) with similar curve progressions for the five investigated wires (Figure 2a,b). An accumulation of the measured values was found at the end of the test sequence since only marginal tooth movement was seen (Figure 2a,b). The initial intrusive forces of Biostarter 0.014″ archwires were almost twice as high as 0.012″ archwires (F_z_ = 1.074 N), and the force curves deviated further from each other (Figure 2c). Yet, similar curve shapes were seen for the protruding moments of Biostarter 0.014″ archwires with a mean initial moment of M_x_ = −4.753 Nmm and a cluster toward the end of the measurement, indicating the short-term inversion of the direction of motion (Figure 2d). The initial intrusive forces of Biostarter 0.016″ archwires (F_z_ = 1.578 N) were 0.5 N higher than those of 0.014″ archwires and 1 N larger compared to 0.012″ archwires, respectively (Figure 2e). The force curve progressions were similar, with moderate diversification and greater flattening with a cumulation of values toward the end (Figure 2e). Mean protruding moments of M_x_ = −7.261 Nmm were found for Biostarter 0.016″ archwires, and the curve shapes were steeper with distinct curvatures and a clustering of data at the end (Figure 2f).

### 3.2. Comparison of Tooth Intrusion Results with 0.012″, 0.014″ and 0.016″ NiTi Archwires without Intrusion Steps

Considering 0.012″ NiTi archwires, no significant differences were found between wires by different manufacturers regarding initial intrusive forces and the gradient of the initial forces along the reference distance of 0.8 mm (Figure 3a). However, AlignXF 0.012″ showed significantly higher initial protruding moments compared to Biostarter 0.012″ (*p* = 0.004) as well as the greater ascent of protruding moments (*p* = 0.022) (Figure 3b). These initial differences of around 1 Nmm converged toward the end of the reference distance (Figure 3b).

Regarding 0.014″ NiTi archwires, most wires by different manufacturers presented no differences regarding the initial intrusive forces, and no significant differences were found considering the ascent of initial forces along the reference distance. Only significant lower initial intrusive forces were seen for Sentalloy light 0.014″ compared to both Sentalloy heavy 0.014″ (*p* = 0.006) as well as AlignXF 0.014″ (*p* = 0.005) (Figure 3c). The initial protruding moments of AlignXF 0.014″ archwires were significantly higher compared to Sentalloy light 0.014″ (*p* = 0.016), Copperloy 0.014″ (*p* = 0.013) and Biostarter 0.014″ (*p* = 0.002) (Figure 3d). Considering the gradient of the protruding moments, some couples of investigated 0.014″ wires were largely congruent along the reference distance, e.g., AlignXF 0.014″ and Sentalloy 0.014″ heavy. Biostarter 0.014″, Copperloy 0.014″ and Sentalloy 0.014″ presented similarities at the beginning of the experiment; however, these varied toward the end. Some wires presented significant differences considering the ascent of protruding moments, e.g., Copperloy 0.014″ and AlignXF 0.014″ (*p* = 0.002) as well as Biostarter 0.014″ and AlignXF 0.014″ (*p* = 0.008).

As for 0.016″ NiTi archwires by different manufacturers, significantly lower initial intrusive forces were solely found for Sentalloy light 0.016″ compared to AlignXF 0.016″ (*p* = 0.032) and Sentalloy heavy 0.016″ (*p* = 0.013) (Figure 3e). Considering the gradient of initial intrusive forces along the reference line, most wires presented no differences. However, significant differences were seen comparing Biostarter 0.016″ with AlignXF 0.016″ (*p* = 0.034) and Sentalloy heavy 0.016″ (*p* = 0.016) depicted by the force curve of Biostarter 0.016″ intersecting those of AlignXF 0.016″ and Sentalloy heavy 0.016″ after a 0.65 mm intrusion distance (Figure 3e). Greater variety was found regarding the initial protruding moment and its gradient along the reference line. Align XF presented the highest initial protruding moment (M_x_ = −11,466 Nmm), which was significantly higher compared to Biostarter 0.016″ (*p* = 0.024), Copperloy 0.016″ (*p* = 0.004), and Sentalloy light 0.016″ (*p* = 0.004). The curves of the protruding moments presented great differences up to 5 Nmm at the beginning and converged toward the end (Figure 3f). Significant differences regarding the ascent of the protruding moments were found for AlignXF 0.016″ compared to Biostarter 0.016″ (*p* = 0.007), Copperloy 0.016″ (*p* = 0.034) and Sentalloy light 0.016″ (*p* = 0.038). 

### 3.3. Comparison of Forces and Moments of Straight NiTi Archwires Compared to Archwires with Intrusion Steps

The same measurements as those presented above for straight wires were performed for archwires with intrusion steps; however, this was conducted solely for the following wires of the manufacturer Dentsply Sirona: Sentalloy 0.012″ medium, Sentalloy 0.014″ light, Sentalloy 0.014″ medium, Sentalloy 0.014″ heavy, Sentalloy 0.016″ light, Sentalloy 0.016″ medium and Sentalloy 0.016″ heavy. Overall, archwires with intrusion steps presented higher initial intrusive forces F_z_ and protruding moments M_x_ compared to archwires without steps considering the same wire dimension (Table 1) (Figure 4). The increase in the force ranged from 0.3 N (Sentalloy 0.014″ light) up to 1.2 N (Sentalloy 0.016″ medium). Notably, the highest increase in the protruding moment was found for Sentalloy 0.016″ medium with intrusion steps (M_x_ = −19.437 Nmm), which were twice as high as Sentalloy 0.016″ medium without steps (M_x_ = −9.609 Nmm). Notably, the modification of NiTi archwires with intrusion steps was accompanied by greater intrusion distances z compared to NiTi archwires without steps (Table 1, Figure 4). The achieved intrusion distances z for all measured NiTi archwires with intrusion steps ranged between 2.205 mm and 2.620 mm and were even higher than the aspired intrusion distance of 1.6 mm (Table 1).

According to straight wires without steps, archwires with intrusion steps showed greater initial intrusive forces with an increasing wire dimension. Furthermore, a higher configuration as declared by the manufacturer (‘heavy’ vs. ‘medium’ vs. ‘light’) regarding the same wire dimension was accompanied by an increase in the force and moment in most cases (Table 1 and Table 2). Sentalloy heavy 0.014″ showed significantly greater initial forces and moments compared to Sentalloy light 0.014″ (F_z_: *p* = 0.001; M_x_: *p* = 0.027) as well as Sentalloy medium 0.016″ compared to Sentalloy light 0.016″ archwires (F_z_: *p* = 0.04; M_x_: *p* = 0.003), while Sentalloy heavy 0.016″ solely presented significantly higher initial forces compared to Sentalloy light 0.016″ (*p* = 0.014). Interestingly, archwires with a smaller wire dimension but higher configuration (‘heavy’ vs. ‘medium’ vs. ‘light’) sometimes showed greater forces or moments compared to archwires with larger wire dimensions yet a lower configuration: Sentalloy 0.014″ in configuration ‘heavy’ presented higher initial forces and moments compared to Sentalloy 0.016″ light (F_z_ = 2.102 N vs. 2.029 N; M_x_ = −13.731 Nmm vs. −10.235 Nmm) and the Sentalloy 0.012″ medium showed higher initial moments compared to Sentalloy 0.014 light (M_x_ = −7.264 Nmm vs. −7.083 Nmm). The highest protruding moment for the measured archwires with intrusion steps was found for the Sentalloy 0.016″ ‘medium’ (M_x_ = −19.437 Nmm) and was even larger than compared to Sentalloy 0.016″ ‘heavy’ (M_x_ = −15.887 Nmm). This might be explained by the great standard deviation from the initial protruding moment of the Sentalloy 0.016 ‘medium’ (±6.198 Nmm) and variations between the configuration as declared by the manufacturer (‘heavy’ vs. ‘medium’ vs. ‘light’) and the actual force/moment output by the archwire.

Compared to NiTi archwires without steps presenting initial intrusive forces F_z_ between 0.56 N (Biostarter 0.012″) and 2.25 N (Sentalloy 0.016″ heavy), NiTi archwires with intrusion steps presented intrusive forces F_z_ between 1.07 N (Sentalloy 0.012″ medium) and 2.98 N (Sentalloy 0.016″ heavy) (Table 2). Regarding the force and moment gradient, forces and moments decreased during the dynamic intrusion of the test tooth regardless of a modification with intrusion steps (Table 2). 

## 4. Discussion

In the present study, forces and moments during simulated initial orthodontic tooth intrusion were investigated using the force-controlled biomechanical test standard ROSS. The overarching goal of the present investigations was to compare NiTi archwires from different manufacturers, material compositions, and diameters in order to identify archwires that provide suitable forces and moments. Furthermore, the effect of intrusion steps of 1.2 mm on the modulation of the force was studied. 

As for the methodology, the focus of the experimental setup was the establishment of a self-regulating dynamic test stand based on a robotic system. Extensive program development work was performed to enable this automatic test setup comprising, e.g., the identification of ‘feedback parameters’. This included the allocation of a movement amplitude to the attached forces and moments, which was used by the robot to move the tooth in order to reduce forces and moments most effectively. As the tooth morphology determines the resistance to the movement, these feedback parameters must be evaluated for each tooth type. The parameters found in our test stand might be helpful for future studies using this software for the advancement of this type of experimental setup. An upper incisor was chosen for clinical reasons as this tooth type was found to have an enhanced risk for OIIRR [13,14,15,16], and upper incisors often require vertical levelling at the beginning of the treatment, e.g., in deep bite cases with elongated upper incisors [1]. Thermal treatment was used for the modification of wires with intrusion steps at a height of 1.2 mm. This height was chosen since a minimal value of 1.2 mm could be achieved in clinical practice using pliers and heat treatment by, e.g., using the Memory-Maker^TM^ (Forestadent, Pforzheim, Germany) [26]. The archwires were manually positioned into the stainless steel intrusion tool, and the lid was bolted afterward according to a published technique [27]. The height of the intrusion step was subsequently verified using a caliper. To enable the comparability of data and data points for each wire, a reference distance of 0.8 mm (50% of the intrusion distance) was chosen. The averaged curves of the measured five samples for each wire with superimposed best-fit straight lines were chosen to allow better comparability of the gradient. After an initial drop, the slope of the force, as well as the moment curves, presented a rather linear progression (Figure 2). The initial drop for the first movement interval (0.5 mm) is a typical irregularity, as found in other biomechanical simulation setups [46]. Moreover, the superimposed best-fit straight lines suggest a rather linear correlation between the force, moment, and intrusion distance, which is only applicable to a limited degree. 

Regarding initial forces, we found that 0.012″ NiTi archwires without steps exerted intrusive forces ranging from 0.561 N to 0.690 N and showed no significant differences regarding the manufacturers. The modification of 0.012″ NiTi archwires with a 1.2 mm intrusion step increased the intrusive force up to 1.066 N. Regarding 0.014″ NiTi archwires, intrusive forces ranging from 0.996 N (Sentalloy light 0.014″, Dentsply Sirona) to 1.321 N (AlignXF 0.014″, Ormco) were found with significant differences between the manufacturers. The modification of 0.014″ NiTi archwires enhanced initial forces up to 1.319 N (Sentalloy light 0.014″, Dentsply Sirona) and 2.102 N (Sentalloy heavy 0.014″, Dentsply Sirona). Whereas most recent studies investigating orthodontic forces are not comparable with our results due to their biomechanical setup [32,33,34,41,47], Perrey et al. detected initial forces of up to 3.5 N ranging between 0.2 N and 3.5 N for 0.014″ archwires and a 0.022″ slot [41], which is partly in accordance with our results, yet, with a greater range. Considering 0.016″ NiTi archwires, we detected initial forces ranging from 1.442 N (Sentalloy light 0.016″, Dentsply Sirona) to 2.254 N (Sentalloy heavy 0.016″, Dentsply Sirona) with significant differences between manufacturers. In accordance with a previous study using ROSS, it was found that 0.016″ NiTi archwires produced excessive force when applied for the levelling of a tooth extruded by 1.6 mm [24]. Step bends applied in 0.016″ NiTi archwires led to force levels between 2.029 N (Sentalloy light 0.016″, Dentsply Sirona) and 2.981 N (Sentalloy heavy 0.016″, Dentsply Sirona), which could lead to severe OIIRR when applied clinically [22].

In our study, the force was applied anterior to the centre of resistance, resulting in a protruding moment around the x-axis, which was accompanied by a palatal root torque [1,31]. The initial protruding moments M_x_ measured in our study ranged between −2.65 Nmm (Biostarter 0.012″, Forestadent) and −11.466 Nmm (AlignXF 0.016″, Ormco) for unmodified NiTi archwires. The lowest moment values were seen for 0.012″ NiTi archwires ranging from −2.65 Nmm (Biostarter 0.012″, Forestadent) to −3.922 Nmm (AlignXF 0.012″, Ormco). A 0.014″ wire dimension showed higher values from −4.753 Nmm to (Biostarter 0.014″, Forestadent) to −7.384 Nmm (AlignXF 0.014″, Ormco) and 0.016″ presented a range from −5.556 (Copperloy 0.016″, Dentsply Sirona) up to −11.466 Nmm (AlignXF 0.016″, Ormco). The moment levels detected by Perrey et al. (range between 8 Nmm and 35 Nmm for 0.014″ archwires and a 0.022″ slot [41]) were significantly higher than our results—even 0.016″ archwires did not reach 35 Nmm in our study. NiTi archwires with intrusion steps presented even greater values ranging between −7.26 Nmm (Sentalloy 0.012″ medium, Dentsply Sirona) and −15.89 Nmm (Sentalloy 0.016″ heavy, Dentsply Sirona). An increased risk for OIIRR was also found when continuous moments between 3 Nmm and 6 Nmm led to the capillary blood pressure being transcended when applied during orthodontic treatment [48]. 

During the simulated intrusion, none of the investigated archwires without steps, regardless of the manufacturer and respective diameter, reached the full intrusion distance of 1.6 mm (archwires without intrusion steps). NiTi archwires with a smaller wire dimension generally reached slightly lower intrusion distances compared to larger wire dimensions, e.g., Sentalloy 0.014 light achieved 59% of the aspired intrusion distance and Sentalloy 0.016″ light reached 71%, respectively (Figure 5). Three exceptions were noted when comparing the archwires of different material compositions, e.g., the Sentalloy 0.014″ light produced lower forces than the Sentalloy 0.012″ medium, and the Sentalloy 0.016″ medium produced lower forces than the 0.014″ Sentalloy heavy (Figure 5). All measured archwires with intrusion steps of 1.2 mm exceeded the intrusion distance of 1.6 mm, e.g., the Sentalloy 0.012″medium with IS achieved 138% and the Sentalloy 0.016″ medium with IS reached 168%, respectively (Figure 5). Adding the intrusion step height to the intrusion distance, a maximum intrusion distance of 2.8 mm can be expected for archwires with intrusion steps; yet, while all analysed archwires with steps exceeded 1.2 mm, none of them reached a 2.8 mm intrusion distance (Figure 5). 

The reason that the aspired intrusion distance was not reached by any investigated archwire can be explained by the results of this study, as the forces already approached zero at lower intrusion distances. For example, the intrusive force F_z_ of the Biostarter 0.012″ archwire approached 0 N after an intrusion distance of only approximately 1 mm (Figure 2). The force control of ROSS is programmed such that the robot reduces the applied forces and moments made by motions in the sense of a simulated tooth movement. As the applied forces approached zero, the amount of the remaining force vector became so small that the biomechanical setup was unable to cause a further reduction in the force, which also explains the noise or the sudden occurrence of enlarged value fluctuations in this range (Figure 2). This result corresponds to clinical observations during orthodontic tooth movement, as archwires of small diameters (e.g., 0.012″) cannot level all vertical deviations, and, therefore, clinicians subsequently use higher archwire dimensions for further vertical alignment [49]. However, the “intrusion efficiency”, i.e., the amount of possible intrusion distance vs. the F_z_ of smaller wire dimensions, is significantly higher compared to larger wire dimensions in general (Figure 6). 

From Figure 6, it is clear that across more or less all suppliers, smaller diameters still deliver significant intrusion distances even though the amount of intrusion forces are only in the range of 25–30% compared to the larger-sized wires. The yellow columns in Figure 6 indicate this conclusion, and the tendency of the “intrusion efficiency” is very clear. This finding may be attributed to the increased play in the bracket slot, thus leading to a reduced frictional force between the archwire and slot [50,51]. For clinical use, this means that smaller wire sizes with delivered intrusive forces far below 1 N may still lead to sufficient intrusion results by reducing the risk of root resorption.

The application of an intrusion step modulated the forces, resulting in greater intrusion distances z (Figure 5). In addition, the disproportion between the slot size and the archwire dimensions may have influenced the achieved intrusion distances. In the present study, NiTi archwires with dimensions of 0.012″/0.014″/0.016″ were investigated with a standard bracket with a slot size of 0.022″. However, the difference in play between 0.012″ and 0.016″ archwires is solely 0.05 mm; hence, the impact of the wire play is probably irrelevant with respect to the dimensions studied. Another aspect to be considered is the effect of the applied thermal treatment. Even though the heat treatment has been conducted under sufficiently controlled conditions using a stainless steel tool with an integrated temperature sensor and an air convection furnace with only minimal temperature gradients, the original manufacturers’ thermomechanical treatment of the NiTi material may have been disturbed, and, therefore, the force-deflection behaviour in the bending of the heat-treated archwires could differ from the as- supplied material [27]. Even though clinically not applicable, the chosen shape-setting process is still considered the most appropriate as it delivers the highest possible reproducibility of all the available shape-setting methods. 

Another interesting material aspect concluded from the present data is that the addition of copper in NiTi alloys (Copperloy 0.014″ and 0.016″) does not reduce forces and moments to the expected extent. It is well known from the literature that Cu reduces the plateau stresses of NiTi alloys by changing the sequence of the martensitic transformation from B2->R > B19′ (monoclinic) toward B2->B19 (orthorhombic) [52] leading to reduced plateau stresses combined with narrower stress hysteresis. In the case of the presented research, the effect of the thermomechanical treatment of binary NiTi seems to be more or less equivalent to alloying with Cu: the 0.014″ Copperloy wire delivers about identical F_z_ values as the binary Sentalloy medium of the same dimension (1.126 N vs. 1.175 N). Furthermore, the 0.016″ Sentalloy medium shows even lower forces compared to the Copperloy wire (1.637 N vs. 1.853 N), giving rise to the conclusion that the thermomechanical treatment of binary NiTi may be as efficient for the adjustment of plateau stresses as alloying with Cu. 

Considering the limitations of this study, the brackets were positioned manually, and the tooth was placed into the physiological position in the Kavo Typodont using a silicone tray, both depending on the practitioner’s skills with possible tolerances. For future trials, digital positioning trays are being considered. Furthermore, the results of the present study were presented as data acting in the centre of resistance (CoR), which was defined as a static point in this experimental setup. However, it was shown that, clinically, the changes in CoR during tooth movement in a clinical in vivo situation depend on tooth morphology, the periodontal ligament, and attached alveolar bone level [1,53]. Further, the CoRot changes during tooth movement depending on the distance between the applied force and the CoR [54,55,56]. Future in vitro trials could consider the dynamic relocation of the CoR and CoRot. Even though the results are not directly transferrable to the clinical in vivo situation due to the missing periodontal ligament and alveolar bone, some relevant conclusions are being delivered.

Regarding clinical implementations, the application of heavy forces during tooth intrusion can lead to apical compression with an increased risk of OIIRR. It was shown that the intrusion of single-rooted premolars did not lead to severe OIIRR when continuous forces of 50 cN were applied for 4 weeks. By contrast, the application of continuous intrusive forces of 100 cN for 4 weeks was accompanied by severe OIIRR, especially in the apical region of the root [22]. Intrusive forces no greater than 0.5 N were recommended for the intrusion of single-rooted teeth [1]. Taken together, unmodified 0.012″ NiTi archwires present ideal initial intrusive forces and moments when applied to the vertical aligning of extruded single-rooted teeth. 0.014″ NiTi archwires are clinically applicable for levelling and aligning when not applied to excessively extruded single-rooted teeth. NiTi archwires with intrusion steps even achieved an overcorrection with greater achieved intrusion distances (>1.6 mm) compared to NiTi archwires without steps not reaching the full intrusion distance. This overcorrection is a consequence of significantly higher forces F_z_ and moments M_x_ exerted by NiTi archwires with steps compared to unmodified NiTi archwires. Hence, the modification of wires with steps could be used to increase the transmission of the force and moments of orthodontic-fixed appliances, e.g., for vertical overcompensation when a larger diameter archwire cannot be placed due to the position of other teeth. Due to the high forces exerted by NiTi archwires with IS, the use of steps should be considered carefully to avoid OIIRR or a loss of anchorage [1,57,58,59,60]. Taken together, intrusion steps should solely be applied in initial levelling archwires, i.e., 0.012″ NiTi archwires for the intrusion of lower incisors and 0.014″ NiTi archwires for upper incisors. 

## 5. Conclusions

The measured intrusive forces and protruding moments during the vertical alignment of an extruded tooth (1.6 mm) exceeded the limit values recommended in the literature to avoid OIIRR for all 0.016″ NiTi archwires and for some measured 0.014″ NiTi archwires. Even simulated tooth intrusion with 0.012″ NiTi archwires presented forces and moments close to the limit value for OIIRR. Hence, a critical evaluation is recommended as to whether NiTi archwires with dimensions greater than 0.012″ should be applied for initial orthodontic tooth intrusion. Overall, increasing the wire dimension was accompanied by increasing the force and moment levels. Furthermore, differences were found regarding 0.014″ and 0.016″ archwires regarding different manufacturers (but not for 0.012″ dimensions). Sentalloy light 0.014″/0.016″ from Dentsply Sirona presented significantly smaller intrusive forces and protruding moments compared to AlignXF 0.014″/0.016″ from Ormco. Biostarter 0.016″ from Forestadent showed the highest intrusion distance accompanied by the second lowest F_z_ compared to other 0.016″ archwires found; yet, these force levels are still too high for the vertical alignment of extruded teeth. The choice of the manufacturer seems to be of subordinate importance regarding smaller wire dimensions, yet might gain importance when greater wire dimensions are used. The modification of NiTi archwires with intrusion steps increased and, in some cases, even doubled the measured intrusive forces and protruding moments while leading to an overcompensation of the intrusion distance. Therefore, the modification of archwires with steps can be a therapeutic, effective tool for orthodontic tooth intrusion. Since intrusive forces up to 2.981 N and moments up to −19.437 Nmm were found after the modification of 0.016″ NiTi archwires with intrusion steps, the choice of the wire dimension is crucial. To avoid OIIRR, intrusion steps should be bent in initial levelling archwires, i.e., 0.012″ NiTi archwires, for the intrusion of the extruded single-rooted tooth.

## Figures and Tables

**Figure 1 bioengineering-10-01356-f001:**
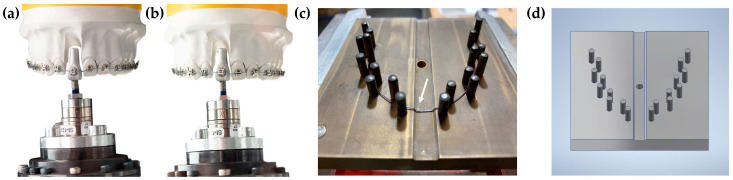
Visualisation of the high-strength titanium alloy Kavo Typodont model adapted to SAM^®^ Axiosplit^®^ mounting plates (SAM Präzisionstechnik GmbH, Gauting, Germany) and the test tooth in (**a**) an extruded position and (**b**) intruded position after orthodontic tooth intrusion using Sentalloy 0.012″ NiTi archwire (Dentsply Sirona, York, PA, USA) with intrusion steps. Graphical presentation of (**c**) the intrusion step template and (**d**) the CAD design for the fabrication of the tool. The counterpart of the shape setting tool is not shown here. The tool temperature was controlled via an internal Ni-CrNi thermocouple located in close proximity to the sample (position is marked by the white arrow in (**c**)).

**Figure 2 bioengineering-10-01356-f002:**
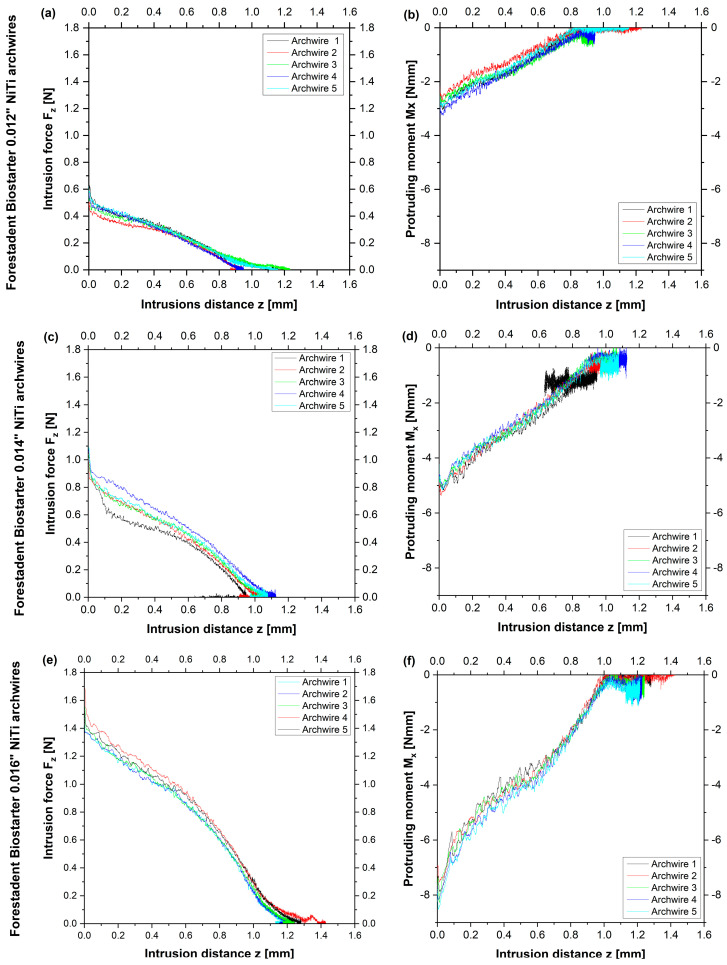
Initial intrusive forces F_z_ [N] in relation to the intrusion distance z [mm] (**a**,**c**,**e**) and the protruding moments M_x_ [Nmm] in relation to the intrusion distance z [mm] (**b**,**d**,**f**) are given for Forestadent Biostarter NiTi archwires without intrusion steps and all measured wire samples. Each sample (1–5) is presented using a colour scheme given next to the figures.

**Figure 3 bioengineering-10-01356-f003:**
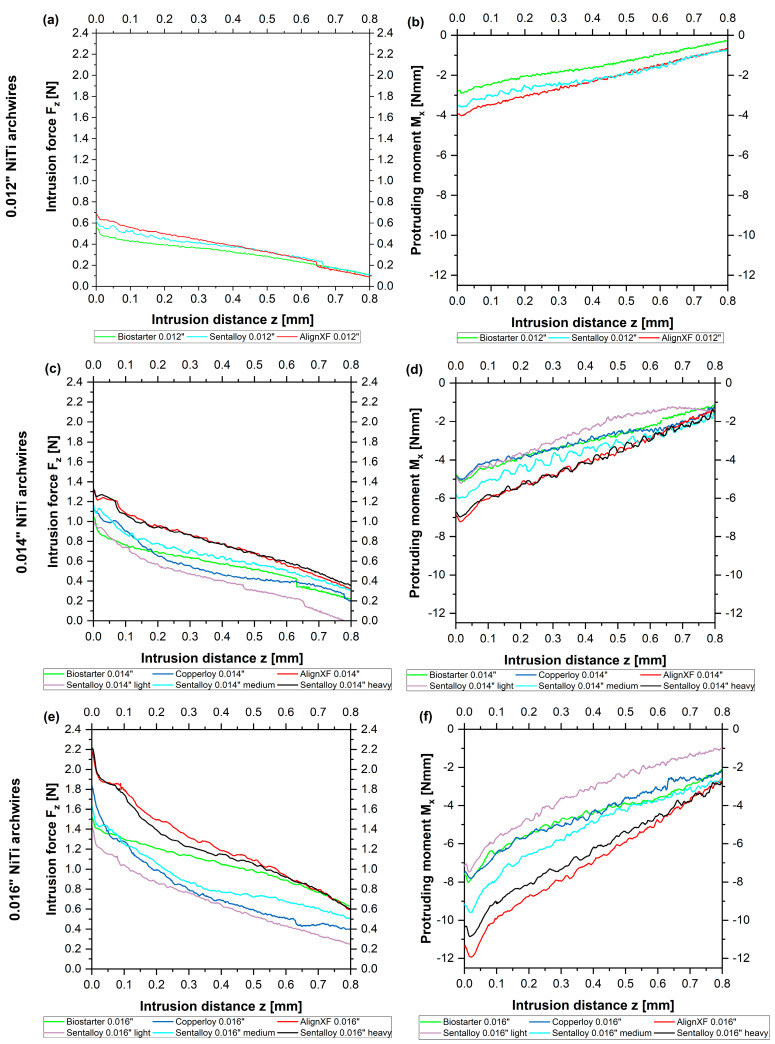
Initial intrusive forces F_z_ [N] (**a**,**c**,**e**) and protruding moments M_x_ [Nmm] (**b**,**d**,**f**) regarding the reference distance of 0.8 mm are given for all measured 0.012″ NiTi archwires and (**a**,**b**) 0.014″ NiTi archwires (**c**,**d**) and 0.016″ NiTi archwires (**e**,**f**) without intrusion steps. The averaged curves with superimposed best fit straight lines are given.

**Figure 4 bioengineering-10-01356-f004:**
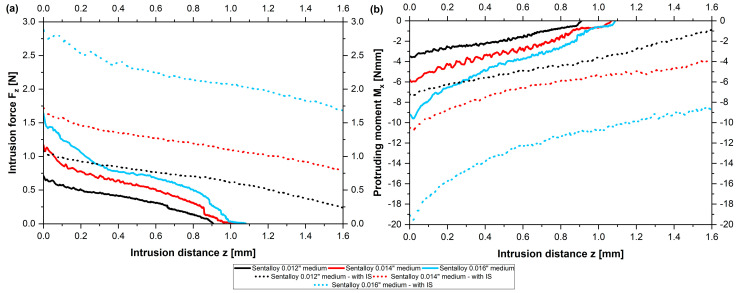
Initial intrusive forces F_z_ [N] in relation to the intrusion distance z [mm] (**a**) and the protruding moments M_x_ [Nmm] in relation to the intrusion distance z [mm] (**b**) are given for Dentsply Sirona Sentalloy 0.012″, 0.014″ and 0.016″ medium NiTi archwires with and without intrusion steps.

**Figure 5 bioengineering-10-01356-f005:**
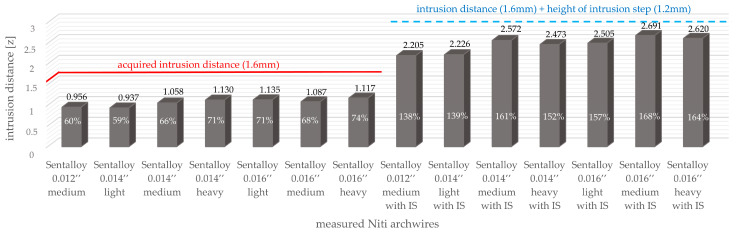
The achieved intrusion distance z [mm] and percentual intrusion [%] is given for all measured NiTi archwires without and with intrusion steps. A red horizontal line is given to visualize the intrusion distance (1.6 mm) and a blue dotted horizontal line visualizes the sum of the intrusion distance and height of the intrusion step.

**Figure 6 bioengineering-10-01356-f006:**
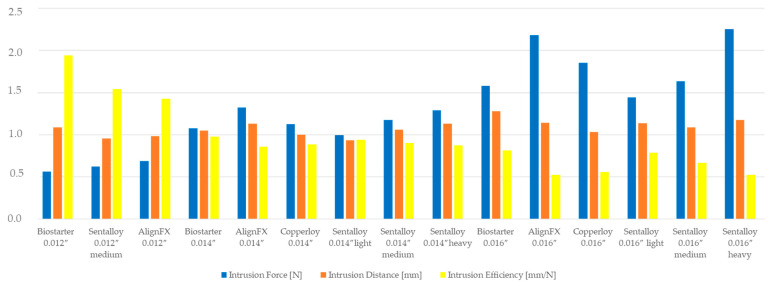
Intrusive force Fz and possible intrusion distance as a function of the wire diameter and manufacturer. Smaller wire diameters deliver smaller forces, but the intrusion distance is in the range compared to larger wire sizes and much higher intrusion forces.

**Table 1 bioengineering-10-01356-t001:** Initial intrusive forces F_z_ [N], initial protruding moments M_x_ [Nmm] and the achieved intrusion distances [mm] are given for all measured straight NiTi archwires and for modified NiTi archwires with intrusion steps.

Manufacturer	Wire and Dimension [Inch]	F_z_ [N]	M_x_ [Nmm]	z [mm]
		MV SD	MV SD	MV SD
Forestadent	Biostarter 0.012″	0.561 ±0.079	−2.649 ±0.268	1.089 ±0.134
	Biostarter 0.014″	1.074 ±0.068	−4.753 ±0.174	1.049 ±0.067
	Biostarter 0.016″	1.578 ±0.105	−7.261 ±0.391	1.281 ±0.085
Ormco	AlignXF 0.012″	0.690 ±0.035	−3.922 ±0.159	0.985 ±0.059
	AlignXF 0.014″	1.321 ±0.103	−7.384 ±0.482	1.133 ±0.040
	AlignXF 0.016″	2.180 ±0.261	−11.466 ±0.634	1.141 ±0.146
Dentsply Sirona	Copperloy 0.014″	1.126 ±0.081	−4.810 ±0.497	0.998 ±0.030
	Copperloy 0.016″	1.853 ±0.506	−5.556 ±3.748	1.031 ±0.086
Dentsply Sirona	Sentalloy 0.012″ medium	0.621 ±0.073	−3.524 ±0.425	0.956 ±0.052
	Sentalloy 0.014″ light	0.996 ±0.071	−5.011 ±0.448	0.937 ±0.186
	Sentalloy 0.014″ medium	1.175 ±0.072	−5.623 ±0.574	1.058 ±0.051
	Sentalloy 0.014″ heavy	1.292 ±0.083	−6.627 ±0.048	1.130 ±0.042
	Sentalloy 0.016″ light	1.442 ±0.127	−6.781 ±0.937	1.135 ±0.172
	Sentalloy 0.016″ medium	1.637 ±0.328	−9.609 ±0.871	1.087 ±0.015
	Sentalloy 0.016″ heavy	2.254 ±0.174	−10.047 ±1.046	1.177 ±0.130
Dentsply Sirona	Sentalloy 0.012″ medium with IS	1.066 ±0.112	−7.264 ±0.857	2.205 ±0.265
	Sentalloy 0.014″ light with IS	1.319 ±0.121	−7.083 ±1.200	2.226 ±0.048
	Sentalloy 0.014″ medium with IS	1.722 ±0.195	−10.342 ±3.985	2.572 ±0.050
	Sentalloy 0.014″ heavy with IS	2.102 ±0.065	−13.731 ±2.884	2.473 ±0.108
	Sentalloy 0.016″ light with IS	2.029 ±0.132	−10.235 ±1.647	2.505 ±0.067
	Sentalloy 0.016″ medium with IS	2.891 ±0.227	−19.437 ±6.198	2.691 ±0.083
	Sentalloy 0.016″ heavy with IS	2.981 ±0.186	−15.887 ±0.891	2.620 ±0.148

MV = mean value, SD = standard deviation, N = Newton, Nmm = Newton millimeter, mm = millimeter, IS = intrusion step.

**Table 2 bioengineering-10-01356-t002:** Presentation of the force and moment gradient along the intrusion reference distance of 0.8 mm for selected unmodified archwires (Biostarter 0.012″, Sentalloy 0.016″ heavy) and NiTi archwires without intrusion steps (Sentalloy 0.012″ medium, Sentalloy 0.016″ heavy). Intrusive forces along the z-axis (F_z_ [N]) and protruding moments along the x-axis are given (M_x_ [Nmm]).

Intrusion Distance [mm]	Biostarter 0.012″	Sentalloy 0.016″ Heavy	Sentalloy 0.012″ Medium with Intrusion Steps	Sentalloy 0.016″ Heavy with Intrusion Steps
	F_z_ [N]	F_z_ [N]	F_z_ [N]	F_z_ [N]
	MV SD	MV SD	MV SD	MV SD
0.000	0.561 ±0.079	2.254 ±0.174	1.066 ±0.112	2.981 ±0.186
0.025	0.475 ±0.066	1.887 ±0.126	1.032 ±0.075	2.775 ±0.110
0.050	0.453 ±0.044	1.860 ±0.105	1.021 ±0.119	2.739 ±0.090
0.075	0.444 ±0.035	1.796 ±0.143	1.023 ±0.080	2.773 ±0.081
0.100	0.416 ±0.036	1.778 ±0.219	0.994 ±0.088	2.818 ±0.065
0.150	0.406 ±0.045	1.508 ±0.142	0.953 ±0.091	2.705 ±0.139
0.200	0.384 ±0.019	1.396 ±0.095	0.933 ±0.072	2.584 ±0.132
0.400	0.323 ±0.026	1.148 ±0.056	0.837 ±0.084	2.411 ±0.091
0.800	0.133 ±0.016	0.607 ±0.030	0.686 ±0.111	2.180 ±0.104
distance [mm]	M_x_ [Nmm]	M_x_ [Nmm]	M_x_ [Nmm]	M_x_ [Nmm]
	MV SD	MV SD	MV SD	MV SD
0.000	−2.649 ±0.268	−11.466 ±0.634	−7.264 ±0.857	−15.887 ±0.891
0.025	−2.755 ±0.465	−11.918 ±0.427	−7.266 ±0.932	−16.832 ±0.554
0.050	−2.599 ±0.517	−11.122 ±0.455	−7.055 ±0.565	−16.263 ±0.670
0.075	−2.552 ±0.265	−10.286 ±0.331	−6.672 ±0.839	−15.128 ±0.968
0.100	−2.570 ±0.419	−9.584 ±0.801	−6.562 ±0.982	−14.046 ±1.484
0.150	−2.362 ±0.322	−9.470 ±0.224	−6.189 ±0.690	−14.843 ±0.417
0.200	−2.061 ±0.245	−8.310 ±0.814	−6.394 ±0.756	−14.253 ±1.512
0.400	−1.773 ±0.285	−7.226 ±0.664	−5.781 ±1.024	−12.222 ±0.960
0.800	−0.141 ±0.126	−2.604 ±0.851	−4.326 ±0.973	−10.722 ±0.716

## Data Availability

The data presented in this study are available on request from the corresponding author. The data are not publicly available.

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
