# Peer review of "Investigation of Forces and Moments during Orthodontic Tooth Intrusion Using Robot Orthodontic Measurement and Simulation System (ROSS)"

_bioengineering, 2023, doi:10.3390/bioengineering10121356_

Round 1
Reviewer 1 Report
Comments and Suggestions for Authors
The article should have better-quality figures.
Author Response
We increased the quality of all figures. All figures have a resolution of
600 dpi.

Reviewer 2 Report
Comments and Suggestions for Authors
It is a nice study conducted and presented well.
Author Response
Thank you very much for this positive review.

Reviewer 3 Report
Comments and Suggestions for Authors
The work concerns a topic of interest in orthodontics and is therefore
of great importance for practitioners in this field.
The scientific investigation has been appropriately carried out with care and in detail. The results obtained are clear and systematically
reported. The conclusions are appropriate and of high interest to those
works in this field. I would not suggest any changes and would recommend the publication of the article in its current state.
Author Response

(The authors gave the same response as above.)
